# Knowledge, attitude, and practice of vasoactive agents infusions: Development and psychometric properties of a questionnaire with chinese clinical nurses

Yanfang Huang[1,2], Yi Chen[1], Yaping Fang[1‡], Junjie Mou[1‡], Shiqin Li[1], Xianying Lei[1¤a], Yuxin Li[3], Zhongping Ai[1,2*¤b]

1 Department of Critical Care Medicine, The Affiliated Hospital of Southwest Medical University, Luzhou, Sichuan Province, China, 2 Nursing Department, The Affiliated Hospital of Southwest Medical University, Luzhou, Sichuan Province, China, 3 Faculty of Nursing, Southwest Medical University, Luzhou, Sichuan Province, China

☯ These authors contributed equally to this work.
‡ YF and JM also contributed equally to this work.
¤aCurrent Address:Department of Critical Care Medicine, The Affiliated Hospital of Southwest Medical University, Luzhou, Sichuan Province, China.
¤bCurrent Address:Nursing Department, The Affiliated Hospital of Southwest Medical University, Luzhou, Sichuan Province,China.
* 407325659@qq.com

## Abstract

### Background

Inconsistencies with guidelines or standards regarding nurses' practice of vasoactive agent infusion have been documented. Adequate knowledge and positive attitudes are critical for compliance. However, there are currently no validated tools specifically designed to measure the knowledge and attitudes related to vasoactive agent infusions among nurses.

### Objective

The aim of this study was to develop and test the validity and reliability of the Chinese mainland version of Knowledge, Attitude, and Practice of Vasoactive Agents Infusions Questionnaire among nurses.

### Methods

The initial questionnaire items were developed through a comprehensive literature review, expert consultation, and pilot study. From February to June 2024, cross-sectional data were collected using convenience sampling from 538 nurses across 9 hospitals in Sichuan Province, China. The reliability and validity of the scale were evaluated through internal consistency reliability, inter-rater reliability, exploratory factor analysis, and confirmatory factor analysis.

**Data availability statement:** All relevant data are within the manuscript and its Supporting Information files.

**Funding:** This work was supported by Sichuan Science and Technology Program2022YFS0632 awarded to X.L. and the Southwest Medical University Project 2022QN027 awarded to Y.H.

**Competing interests:** The authors have declared that no competing interests exist.

## Results

The final version of the questionnaire included 33 items across 3 dimensions, explaining 78.04% of the variance, with item loadings ranging from 0.56 to 0.89. The content validity index ranged from 0.91 to 1.00, and the scale-level content validity index was 0.98.The overall Cronbach's α for the questionnaire was 0.96, with Cronbach's α for each dimension ranging from 0.96 to 0.98. The test-retest reliability for the entire questionnaire was 0.90, and for each dimension, it ranged from 0.90 to 0.94 ($p < 0.05$). The confirmatory factor analysis demonstrated acceptable fit indices for the three-dimensional model: $\chi^2/(df) = 1.135$, $p = 0.200$, RMSEA $= 0.024$, CFI $= 0.986$, TLI $= 0.985$, GFI $= 0.884$, and NFI $= 0.895$.

## Conclusion

The Knowledge, Attitude, and Practice of Vasoactive Agents Infusions Questionnaire demonstrates good reliability and validity, making it a reliable measurement tool for assessing nurses' attitudes and knowledge related to vasoactive agents' infusions. This version will facilitate further research and advancements in this specific field of study.

## 1. Introduction

Vasoactive agents are medications that regulate blood vessel tone and blood flow. By targeting vascular smooth muscle and endothelial cells, they modulate vasoconstriction or vasodilation, influencing blood pressure and organ perfusion [1]. Between 10% and 54% of patients admitted to intensive care units (ICUs) have been reported to receive vasoactive agents for conditions such as sepsis, heart failure, and organ failure. Moreover, factors such as the dose and type of vasoactive drug used are associated with patient mortality. For example, Septic shock patients with a <−50% vasoactive-inotropic score (VIS) response had significantly higher ICU mortality (HR 2.07, 95% CI 1.61–2.66, p < 0.001) compared to those with a ≥ 50% response, especially after receiving vasoactive-inotropic therapy for more than 24 hours [2–4]

Nurses are entrusted with all aspects of medication management, including the preparation, initiation, administration, titration, weaning, and documentation of vasoactive agents. Given the complexity of these medications, nurses face substantial challenges in making dynamic clinical decisions when administering vasoactive agents to patients [5]. A recent observational study in Australia found that nurses often maintain patients' Mean Arterial Pressure (MAP) above the recommended target range of 60 mmHg to >75 mmHg to prevent hypotension. However, this approach frequently overlooks the potential adverse effects of high doses of vasoactive agents [6]. Similarly, studies [7,8]in China have demonstrated the feasibility and safety of peripheral infusion of vasoactive agents with the Chinese Nursing Association establishing clinical guidelines for their management [9]. Despite this, adherence to these standards remains inconsistent. Integrating evidence-based guidelines into clinical

practice remains a significant challenge. As highlighted in a systematic review, nurses in intensive care units worldwide are tasked with managing vasoactive medications, yet the practices employed vary significantly [10].

As summarized in a systematic review, nurses working in intensive care units globally are responsible for managing vasoactive medications. However, practices vary widely, particularly in areas such as dosing adjustments and patient monitoring,

There are several challenges when nurses·adhering to guidelines for vasoactive agent administration, including deficiencies in knowledge regarding accurate drug dosage calculations [11]limited recognition of risks such as arrhythmias [12] and organ damage, and difficulties in operating complex equipment like infusion pumps [13].These factors can contribute to an increased risk of medication errors and adverse events in critically ill patients, a particularly vulnerable population. Despite the increasing emphasis on the safety administration of vasoactive drugs by nursing professionals in recent years, there remains limited insight into the level of knowledge and attitudes employed by clinical nurses [14]. To achieve optimal outcomes, nursing administrators need to comprehensively understand the knowledge, attitudes, and practices of clinical nurses concerning vasoactive agent infusions. Hence, this study aimed to develop the Knowledge, Attitude, and Practice of Vasoactive Agents Infusions Questionnaire (KAPV-AIQ) for nurses. This tool will help identify weaknesses in practice through questionnaire assessments, guide the development of improvement measures. Consequently, it is expected to improve overall nursing quality, ensure patient safety and treatment efficacy, and aid in formulating more detailed and actionable training programs.

## 2. Materials and methods

The study consisted of the following phases:

### 2.1 Phases1: Instrument development

#### 2.1.1 Specify dimensional structure.
To identify the dimensions of our questionnaire, we applied the Knowledge, Attitude, and Practice (KAP) theoryp [15] as a guiding framework and subsequently, an expert panel reached a consensus after two rounds of detailed discussions on the conceptualization of these dimensions, informed by a comprehensive literature review and the specific context of clinical nursing care in China. Nurses' Knowledge of Vasoactive Agent Infusion refers to the comprehensive understanding and awareness that nurses possess regarding the administration of vasoactive medications. This includes their familiarity with the pharmacological properties, indications, contraindications, dosage calculations, infusion protocols, monitoring requirements, potential adverse effects, and emergency management associated with these potent medications. Nurses' Attitude Towards Vasoactive Agent Infusion encompasses their beliefs, perceptions, and feelings about the administration and management of vasoactive medications. This includes their confidence in handling these potent drugs, their commitment to adhering to safety protocols, their sense of responsibility in monitoring and managing patients receiving such infusions, and their overall perspective on the importance and impact of vasoactive agent administration on patient outcomes. Nurses' Practice of Vasoactive Agent Infusion refers to the actions and procedures that nurses perform in the administration of vasoactive medications. This involves the correct preparation, dosage calculation, and administration of these drugs, continuous patient monitoring, prompt identification and management of adverse reactions, adherence to clinical protocols and guidelines, and accurate documentation of the infusion process.

#### 2.1.2 Item generation.
A literature review was conducted using databases such as PubMed, Cochrane Library, Embase, and Scopus to identify relevant studies on the administration of vasoactive drugs. The search included key terms such as "vasoactive drugs" "vasopressors" "infusion care" "intravenous infusion" "medication administration" "safety management"et al,. The search covered the period from the inception of each database to December 2023. After an initial screening by two independent researchers, the final selection of literature was made through consensus, which included two guidelines [16,17], two expert consensuses [18–19], one evidence summary [11], the Health of the People's Republic

of China industry standards (WS/T 433–2023), and the Chinese Nursing Association Group Standard (T/CNAS 22–2021). These sources formed the basis for developing a preliminary questionnaire with 43 items across three dimensions: knowledge, attitude, and practice.

**2.1.3 Item screening.** In this study, 239 valid questionnaires were used for factor analysis to screen items, with correlation coefficient analysis and factor analysis applied. Items were retained if their correlation with the total scale score (r > 0.4) was statistically significant. Factor analysis was conducted to extract factor loadings, with items deleted if their factor loading was ≤ 0.5 or communality ≤ 0.2. Internal consistency was assessed using Cronbach's α, and items that reduced α when deleted were removed. Subsequently,in the Knowledge and Behavior dimension, three items were removed based on their low correlation with the total score. After factor analysis, seven items with factor loadings below 0.4 were also deleted (See S1 File).The remaining 33 items had correlation coefficients ranging from 0.557 to 0.904 with the total questionnaire score, all exceeding 0.4 and showing statistical significance (P < 0.001) (See Table 1).

**2.1.4 Pilot survey.** In February 2024, a pilot survey was conducted using convenience sampling to select nurses from a tertiary teaching hospital in Sichuan Province. The survey was administered online. Inclusion criteria for participants were possession of a valid nursing license, formal employment by the hospital, engaged in frontline clinical nursing work, and voluntary participation in the survey. Exclusion criteria included nurses on sick leave, maternity leave, or away for training purposes. Participants evaluated the questionnaire using a Likert 5-point scale, assessing several key aspects: the rationality of the questionnaire design, the clarity and comprehensibility of the text, feasibility, willingness to self-assess using the questionnaire, comfort level while answering, and willingness to discuss the questionnaire with researchers. Issues and suggestions raised by participants were also collected and documented.

A total of 60 valid questionnaires were returned. All participants found the questionnaire design rational, the text clear and comprehensible, and the questionnaire feasible. They expressed willingness to use the questionnaire for self-assessment, felt comfortable during the answering process, and were willing to discuss the questionnaire content with other researchers. The average time to complete the questionnaire was 218 seconds.

**2.1.5 Content validity evaluation.** In this study, eleven experts, including critical care specialists, nursing experts, and clinical pharmacists, evaluated the content validity, linguistic clarity, and clinical relevance of the questionnaire items. Content validity was assessed using a 4-point Likert scale, where scores ranged from 1 ("not relevant") to 4 ("highly relevant"). The item-level content validity index (I-CVI) was calculated as the proportion of experts rating an item as 3 or 4 out of the total number of experts. The scale-level content validity index/average (S-CVI/Ave) was determined as the percentage of items rated 3 or 4 by all experts, indicating overall consensus. Items with an I-CVI of 0.78 or higher from at least three experts and an S-CVI/Ave of 0.90 or higher were considered to demonstrate acceptable content validity [20].

We collected 11 valid responses. All experts held at least an associate senior professional title and had a master's degree or higher, with an average of 26.00 years of professional experience (SD = 8.83 years). To enhance clarity, the experts recommended the following revisions: changing "adverse reactions of vasoactive agents" to "common adverse reactions of vasoactive agents"; revising "monitoring blood pressure, heart rate, heart rhythm, etc." to "monitoring vital signs"; and updating "selection of infusion routes for vasoactive agents" to "principles for selecting infusion routes for vasoactive agents." Regarding item 7, "Do you know the specifications of vasoactive agents?" one expert suggested its removal due to limited relevance. However, after review and factor analysis (with factor loading > 0.5, see **Table 1**), the item was retained.

## 2.2 Phases2: Test the psychometric properties of the questionnaire

**2.2.1 Design, setting and sample.** A cross-sectional study was conducted from February 2024 to June 2024. Convenience sampling was used to select clinical nurses from 6 tertiary hospitals and 3 secondary hospitals in Sichuan Province. Eligibility and exclusion criteria were consistent with the pilot survey. The instruments comprised a general information questionnaire (covering age, gender, education, title, and years of clinical experience et al.) and the

**Table 1. Exploratory factor analysis structure matrix of Knowledge, Attitude, and Practice of Vasoactive Agents Infusions Questionnaire (N = 239).**

| Items | Factors | | |
|---|---|---|---|
| | **Knowledge** | **Practice** | **Attitude** |
| K1.Definition of vasoactive agent | **.827** | .177 | .194 |
| K2.Classification of vasoactive agent | **.865** | .123 | .173 |
| K3.Common adverse reactions of vasoactive agent | **.856** | .107 | .190 |
| K4.Principles of selecting infusion routes for vasoactive agent | **.855** | .246 | .167 |
| K5.Requirements for infusing vasoactive agent | **.849** | .226 | .112 |
| K6.Proper configuration of vasoactive agent | **.868** | .229 | .076 |
| K7.Specifications of commonly used vasoactive agent | **.898** | .169 | .109 |
| K8.Precautions during the infusion process of commonly used vasoactive agent | **.895** | .231 | .108 |
| K9.Use of pump for infusing vasoactive agent | **.836** | .314 | .128 |
| K10.Correct use of infusion pumps | **.829** | .321 | .091 |
| K11.Common alarms and management of infusion pumps | **.789** | .343 | .113 |
| K12.Labeling requirements for infusion ports when simultaneously infusing multiple vasoactive agent | **.771** | .361 | .122 |
| K13.Conditions appropriate for using the dual-pump method for continuously infusing vasoactive agent | **.836** | .209 | .102 |
| K14.Monitoring frequency of vital signs during initial use, dose adjustment, and stabilization of vasoactive agent | **.832** | .287 | .088 |
| K15.Procedure for stopping vasoactive drug infusion | **.779** | .213 | .157 |
| A4.Recognizing the vital importance of clinical nurses mastering vasoactive agent care knowledge | .194 | .299 | **.845** |
| A2.Emphasizing the vital role of dynamic assessment during vasoactive agent infusion | .195 | .300 | **.872** |
| A3.Commitment to integrating standards of vasoactive agent infusion into clinical practice | .188 | .337 | **.920** |
| A5.Highlighting the importance of regular specialized training on vasoactive agent infusion | .151 | .326 | **.861** |
| A1.The crucial role of guidelines for vasoactive agent infusion in clinical practice | .163 | .332 | **.854** |
| P4.Vigilant monitoring of infusion pathway and surrounding skin conditions is conducted throughout. | .290 | **.678** | .340 |
| P10.Continuous observation of infusion rate and remaining drug volume is maintained. | .296 | **.751** | .321 |
| P5.Each syringe used during infusion is labeled with patient details, drug specifics, and preparation information. | .196 | **.787** | .200 |
| P12.Prompt replacement of the infusion line is ensured upon reflux detection. | .345 | **.700** | .329 |
| P6.Both ends of the infusion line are clearly labeled during setup. | .355 | **.560** | .172 |
| P13.Preparations for continued infusion are made as drug completion nears, ensuring seamless continuation. | .244 | **.824** | .216 |
| P7.Syringe and infusion tubing are replaced when changing medications. | .284 | **.746** | .126 |
| P11.Collaboration with physicians ensures alignment with therapeutic goals during infusion, with ongoing adjustments and monitoring of vital signs. | .231 | **.858** | .159 |
| P1.Prior to vasoactive agent administration, a dual-person check ensures prescription accuracy regarding dose, method, and infusion rate, with immediate physician clarification if needed. | .219 | **.852** | .227 |
| P2.Assessment includes verifying vascular access, infusion pump functionality, and battery status before drug administration. | .244 | **.797** | .212 |
| P8.During connection of infusion lines, ensuring medication flows correctly to the interface is prioritized. | .184 | **.846** | .154 |
| P3.Precise preparation of drug concentration and infusion rate is conducted according to prescription guidelines before starting infusion. | .149 | **.860** | .219 |
| P9.Ongoing assessment of patient vital signs, peripheral circulation, and urine output is integral during drug administration. | .225 | **.857** | .183 |
| Eigenvalues | 18.19 | 5.14 | 2.42 |
| % of Variance Explained | 35.16 | 28.49 | 14.38 |
| Cumulative % of Variance Explained | 35.16 | 63.65 | 78.04 |

KAP-VAIQ. To assess test-retest reliability, 20 respondents were conveniently selected for a follow-up survey two weeks after the initial data collection.

A sample size of at least 330–660 participants was estimated to be sufficient for factor analysis, based on the recommended item-to-response ratio of 10:1–20:1 [21]. This estimate assumes the KAP-VAIQ contains 33 items, excluding the 60 nurses involved in the acceptability evaluation and the 20 nurses used for test-retest reliability.

**2.2.2 Procedures.** From February 6th, 2024, online recruitment advertisements and questionnaire links were disseminated through WeChat and WenJuanXing to invite voluntary participation from nurses at selected hospitals. The advertisements included study details and contact information. Participants were required to review the study introduction and consent form before accessing the questionnaire. Only after providing informed consent could, they proceed with completing the questionnaire. Data were downloaded from the platform after survey completion, ensuring no duplicate responses due to the platform's unique identifier system.

**2.2.3 Internal reliability and test-retest stability.** Cronbach's alpha was used to assess the internal consistency of the questionnaire. A value greater than 0.7 indicates good internal consistency. The test-retest reliability was assessed using the intraclass correlation coefficient (ICC), which ranges from 0 to 1. A coefficient below 0.4 indicates poor reliability, while a value above 0.75 indicates good reliability [22,23].

**2.2.4 Exploratory factor analysis and confirmatory factor analysis.** Exploratory factor analysis (EFA) and confirmatory factor analysis (CFA) were used to examine the construct validity of the questionnaire. EFA helps identify the underlying structure of the data by grouping related items, while CFA is used to confirm whether the data fit the hypothesized structure..Kaiser-Meyer-Olkin (KMO) and Bartlett's test of sphericity were considered to check the adequacy and suitability of the sample data for factor analysis. Principle Component Methods (PC) and Varimax with Kaiser Normalization were used to do the factor extraction to identify meaningful items. KMO value should be more than 0.5 and Bartlett's test had a probability of <0.05 which means that correlational matrix could be factorable [24]. The criterion for factor extraction is to retain factors with eigenvalues greater than 1. The model fit of the SEM was reflected by the fit indices including the model chi-square ($\chi^2$), Goodness of Fit Index, GFI, Root Mean Square Residual RMR, Root Mean Square Error of Approximation, RMSEA, Comparative Fit Index, CFI, Goodness of Fit Index, GFI, Normed Fit Index, NFI, Tucker-Lewis Coefficient, TLI [25].

## 2.3 Ethical consideration

Ethical approval was obtained from Southwest Medical University Hospital, China, with ethical approval codes KY2024064. The informed consent will be posted first on the "Wenjuanxing" web application. Only if they agree will the questionnaires be displayed for them to complete. If they do not agree, they will be directed to "exit" the platform and will receive a "thank you" message. All participants have the right to refuse or withdraw from the study at any time without penalty. All information gathered for this study stored on a password-protected computer in the form of electronic files and kept confidential from those who do not have the right to know.

## 2.4 Data analysis

After the original data is downloaded from "Wenjuanxing" platform, it is sorted and imported into SPSS 28.0 through Excel. The demographic data of the participants, the reliability, and the EFA was described and analyzed using SPSS. The CFA was used the Analysis of Moment Structure (AMOS). In a 1:1 ratio, the initial 239 samples underwent EFA, and the subsequent 239 samples underwent CFA.

## 3. Results

### 3.1 The final version of questionnaire

The final version of the KAP-VAIQ consists of three dimensions: Knowledge with 15 items, Attitudes with 5 items, and Practices with 13 items. The questionnaire can be found in the Table 1. The Knowledge dimension used a 5-point Likert

scale ranging from "Not at all familiar" to "Very familiar" The Attitudes dimension used a 5-point Likert scale ranging from "Strongly disagree" to "Strongly agree". The Practices dimension used a 5-point Likert scale ranging from "Never" to "Always". Scores are assigned from 1 to 5 for each item, with the total questionnaire score ranging from 33 to 165 points. A higher score indicates a greater level of knowledge, attitudes, and practices related to vasoactive agent infusion nursing among the nurses..

### 3.2 Content validity

The questionnaire demonstrated excellent content validity, with an item-level Content Validity Index (I-CVI) ranging from 0.91 to 1.00 and a scale-level Content Validity Index/average (S-CVI/Ave) of 0.98.

### 3.3 Participants' characteristics

A total of 535 nurses participated in the psychometric assessment, with 478 valid questionnaires collected, yielding a response rate of 89.34%. The majority of participants were women (83.24%). The mean age of the participants was 32.86 years (SD = 7.18), with a median age of 32 years. A significant portion (81.63%) held a bachelor's degree or higher. All participants were registered nurses, including 425 (88.91%) from tertiary hospitals and 53 (11.08%) from secondary hospitals. Among them, 52 (10.87%) were internal medicine nurses, 142 (29.70%) were surgical nurses, 189 (39.53%) were critical care nurses, 30 (6.27%) were operating room/anesthesia nurses, 21 (4.39%) were pediatric nurses, and 16 (3.34%) were from other specialties. Regarding professional titles, 253 (52.92%) held junior titles, 193 (40.37%) held intermediate titles, and 32 (6.69%) held senior titles or higher. The average clinical nursing experience was 8.32 years (SD = 5.07).

### 3.4 Reliability

The questionnaire demonstrated excellent internal consistency, with a Cronbach's alpha coefficient of 0.96 for the overall scale and 0.96 to 0.98 for the individual dimensions. The test-retest reliability was also high, with an intraclass correlation coefficient (ICC) of 0.90 for the overall questionnaire, and ICCs for the individual dimensions ranged from 0.90 to 0.94 ($P < 0.05$).

### 3.5 Exploratory factor analysis

Exploratory factor analysis (EFA) was conducted on 239 samples. The Kaiser-Meyer-Olkin (KMO) measure of sampling adequacy was 0.945, indicating that the data were suitable for factor analysis. Bartlett's test of sphericity was significant ($p < 0.001$), further supporting the factor analysis. Using principal component analysis and applying the criterion of eigenvalues greater than 1, three common factors were extracted, accounting for 78.04% of the cumulative variance. The factors' loadings were computed using varimax rotation, revealing that all item loadings were greater than 0.50, with factor loadings ranging from 0.56 to 0.89 on their respective common factors.

### 3.6 Confirmatory factor analysis

The model yields the acceptable indices: $\chi2/df = 1.135$, $p = 0.020$, Goodness of Fit Index, GFI = 0.884, Root Mean Square Residual RMR = 0.030, Root Mean Square Error of Approximation, RMSEA = 0.024, Comparative Fit Index, CFI = 0.986, Goodness of Fit Index, GFI = 0.884, Normed Fit Index, NFI = 0.895, Tucker-Lewis Coefficient, TLI = 0.985.The final model is shown in Fig 1.

## 4. Discussion

In this study, we developed and tested a preliminary Chinese questionnaire to assess the knowledge, attitudes, and practices of clinical nurses concerning vasoactive agent infusions. As far as we know, this type of comprehensive, validated

questionnaire had never been conducted previously. The gap in the literature stems from the fact that most existing measurements of nurses' knowledge and practices related to vasoactive agents were based on empirical research and lacked psychometric validation, which affected the accuracy of assessments and limited their external validity and applicability [26,27]. This gap is critical, as the safe administration of vasoactive agents is contingent upon nurses' precise knowledge and well-founded practices.Our study addresses this gap by offering a reliable and valid instrument, which can inform the design of targeted educational programs and improve clinical practice. By measuring nurses' knowledge and practices quantitatively, this tool enables the development of more effective educational curricula tailored to the specific needs of clinical nurses, thereby enhancing patient safety and care quality in clinical settings [28].

The final version of KAP-VAICQ, three self-assessments dimensions of knowledge with 15 items, attitude with 5 items, and practice with 13 items, were designed at the individual nurse level, which qualities were consistency with the Theory of Knowledge, Attitude, and Practice (KAP). According to the KAP theory, knowledge, attitudes, and practices are interrelated, with knowledge forming the foundation for attitude change, which in turn influences the adoption of practices [29].In the context of this study, the Knowledge dimension assesses nurses' understanding of the physiological mechanisms and safe administration practices for vasoactive agents, while the Attitudes dimension evaluates the nurses' beliefs and perceptions regarding the importance of proper administration. The Practices dimension measures the actual behaviors exhibited by nurses, including adherence to safety protocols and clinical guidelines. This alignment with the KAP theory makes the questionnaire a robust tool for assessing the cognitive and behavioral components that influence nurses' practices in vasoactive agent infusion.

In this study context, it is vital to improve the standardization practices of nurses in the administration of vasoactive agents to improve the safety of patients' medication and the quality of care. Hence, the KAP-VAIQ can help nursing administrators and researchers assess the current state of nurses' knowledge, attitudes, and practices concerning vasoactive agent infusions. Based on the assessment results, nursing administrators can implement targeted initiatives to improve nurses' adherence to the standardized management of vasoactive agents. These initiatives may include increasing relevant training, optimizing clinical workflows, enhancing communication among healthcare teams, and ensuring the availability of necessary resources.Additionally, the questionnaire can serve as a tool for evaluating the effectiveness of training programs over time. Given the complexity of vasoactive agent management, this questionnaire requires further exploration, particularly regarding extravasation. Additionally, studies have shown that factors such as intelligent technology [30] and organizational safety climate [31] can enhance nurses' knowledge through real-time feedback, shape their attitudes toward safety protocols, and improve their practices by providing better resources..

In this study, Cronbach's alpha coefficient and retest reliability method were used to assess the internal and external reliability of the questionnaire. The study results demonstrated strong internal consistency within questionnaire. Retest reliability analysis indicated good repeatability, and content validity assessment confirmed a high alignment between the measured content and the intended objectives. Besides, to assess the consistency and stability of the questionnaire framework, two different sets of data were used for EFA and CFA. In EFA, the Principal Component (PC) methods and Varimax with Kaiser Normalization were chosen as the preliminary solution [32]. After analysis, we obtained 3 factors that explained 35.16%, 28.49%, 14.38% respectively of the variance, and the accumulated percentage of explained variance was 78.04%. According to the evaluation criteria of CFA indicator [33], the results of yields the acceptable indices, indicating a good fit to the theoretical structure of KAP.

## Limitations

There are some limitations in this study. Firstly, convenience sampling was used, and the web-based questionnaire management platform indicated that all respondents' IP addresses were located in one province in western China, which may limit the generalizability of the findings. To address this, future studies could consider using a more diverse sampling approach, such as stratified or random sampling, and expanding the geographic scope to include multiple provinces or regions. Secondly, while we assessed content and construct validity, no criterion validity test was applied to this scale.

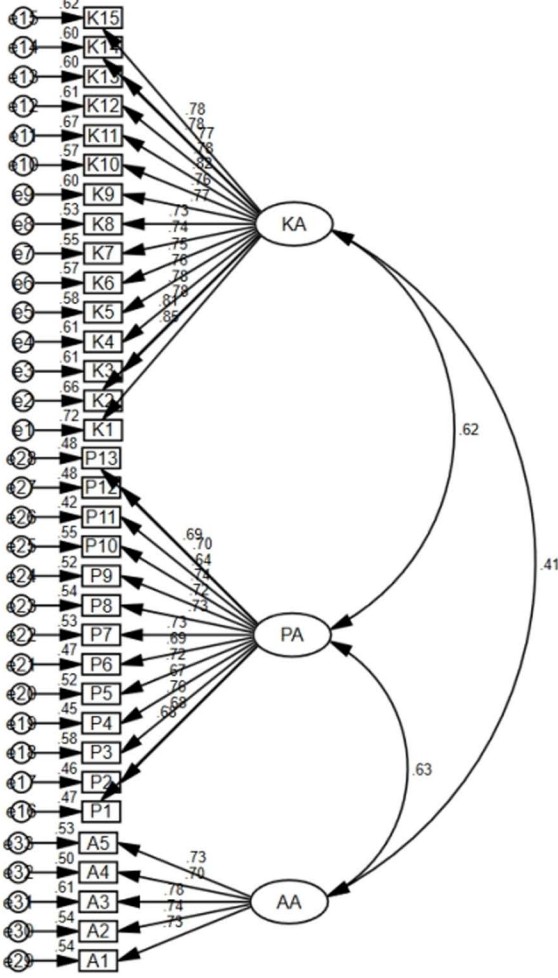

**Fig 1. Confirmatory factor analysis of Knowledge, Attitude, and Practice of Vasoactive Agents Infusions Questionnaire (N = 239).**

Future research should consider testing the criterion validity of the KAP-VAIQ by comparing it with established instruments or clinical outcomes to further validate its accuracy and relevance.

## Conclusions

In conclusion, the KAP-VAIQ has significant implications for improving patient safety and the quality of care in the administration of vasoactive agents. This validated questionnaire can be used to assess and monitor the effectiveness of nursing education and training programs, ensuring that nurses possess the necessary knowledge, attitudes, and practices for safe vasoactive agent infusion. Furthermore, it can be applied in diverse healthcare settings to identify gaps in practice and inform evidence-based interventions. Ultimately, the KAP-VAIQ can play a key role in enhancing the standardization and safety of vasoactive agent administration, contributing to improved patient outcomes.

## Supporting information

**S1 File.  Summary of items deleted during the item screening process.**
(DOCX)

**S2 Data.**
(ZIP)

## Acknowledgments

The authors would like to express their gratitude to all hospital administrators for their support during the data collection process. The authors would also like to thank all the participants in this study.

## Author contributions

**Conceptualization:** Yanfang Huang.

**Data curation:** Yanfang Huang, Yi Chen.

**Formal analysis:** Yanfang Huang, Yi Chen.

**Funding acquisition:** Yanfang Huang, Xianying Lei.

**Investigation:** Yanfang Huang, Yaping Fang, Shiqin Li, Yuxin Li.

**Methodology:** Yanfang Huang, Yaping Fang.

**Resources:** Xianying Lei.

**Software:** Junjie Mou.

**Supervision:** Yanfang Huang, Zhongping Ai Ai.

**Validation:** Junjie Mou.

**Visualization:** Shiqin Li.

**Writing – original draft:** Yanfang Huang, Yi Chen, Yaping Fang.

**Writing – review & editing:** Yanfang Huang, Zhongping Ai Ai.

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
