## [Decision Letter · Decision Letter 0]

29 Dec 2024

PONE-D-24-41867Knowledge, Attitude, and Practice of Vasoactive Agents Infusions Development and Psychometric Properties of a Questionnaire with Chinese Clinical NursesPLOS ONE

Dear Dr. Huang,

Thank you for submitting your manuscript to PLOS ONE. After careful consideration, we feel that it has merit but does not fully meet PLOS ONE’s publication criteria as it currently stands. Therefore, we invite you to submit a revised version of the manuscript that addresses the points raised during the review process.

We look forward to receiving your revised manuscript.

Kind regards,

Elsayed Abdelkreem, MD, PhD

Academic Editor

PLOS ONE

“Supported by Sichuan Science and Technology Program(2022YFS0632).”

Reviewers' comments:

Reviewer's Responses to Questions

**Comments to the Author**

1. Is the manuscript technically sound, and do the data support the conclusions?

Reviewer #1: Yes

Reviewer #2: Yes

Reviewer #3: Yes

2. Has the statistical analysis been performed appropriately and rigorously? 

Reviewer #1: Yes

Reviewer #2: Yes

Reviewer #3: Yes

3. Have the authors made all data underlying the findings in their manuscript fully available?

Reviewer #1: Yes

Reviewer #2: Yes

Reviewer #3: Yes

4. Is the manuscript presented in an intelligible fashion and written in standard English?

Reviewer #1: Yes

Reviewer #2: Yes

Reviewer #3: Yes

5. Review Comments to the Author

Reviewer #1: 1. The authors conducted a rigorous procedure to develop a valid and reliable instrument (KAP-AIQ) to measure the knowledge, attitude and practices related to vasoactive infussion among nurses in mainland china. the data from pilot and main study article were congruent and support the conclusion of this study. Generally, KAP-AIQ has the potential for wide application in clinical nursing research

2.Data analysis was appropriate.

3 the authors lprovide underlying data suppoorting their findings.

4. The manuscript is presented using simple English and easy to read and understand.

Reviewer #2: Dear editor in PLOS ONE

Thank you for your invitation to review manuscript entitled " "

Introduction

Comment 1: Consider rephrasing to "Vasoactive agents are medications that regulate blood vessel tone and blood flow."

Comment 2: "By targeting vascular smooth muscle and endothelial cells, they modulate..."

Comment 3: "Between 10% and 54% of ICU patients receive vasoactive agents for conditions such as sepsis, heart failure, and organ failure."

Comment 4: "Moreover, the dose and type of vasoactive drug are associated with patient mortality."

Comment 5: "Nurses are responsible for all aspects of vasoactive medication management, including..."

Comment 6: "Due to the complexity of vasoactive agents, nurses face challenges in making timely clinical decisions when administering these medications."

Comment 7: "A recent observational study found that Australian nurses often maintain MAP above target levels, potentially leading to adverse effects from high-dose vasoactive agents."

Comment 8: "This poses a significant risk to patient safety, especially for those with hemodynamic instability."

Paragraph 2: Consider combining comments 1 and 2 to improve flow.

Paragraph 3: The transition from the Australian study to the Chinese context could be smoother. Perhaps add a comment like, "While challenges exist globally, the Chinese healthcare system is also facing similar issues."

Paragraph 4: The comment about the Chinese Nursing Association's standard could be rephrased to be more concise.

Comment 4: Consider adding a citation for the mortality association.

Comment 7: Specify the target MAP levels.

Paragraph 4: Explain the specific challenges nurses face in implementing the guidelines

Material and Method

Paragraph 2.1.1: Consider combining the first two sentences to improve flow.

Paragraph 2.1.2: The sentence about the literature review could be simplified. For example, "A literature review was conducted using databases like PubMed, Cochrane Library, Embase, and Scopus to identify relevant studies on vasoactive drug administration."

Paragraph 2.1.3: The explanation of the correlation coefficient analysis and factor analysis could be simplified. Consider using a table to summarize the key criteria for item selection.

Paragraph 2.1.3: The specific items that were excluded could be listed in a table for better readability.

Paragraph 2.1.5: Consider rephrasing the sentence about the expert evaluation to: "Eleven experts, including critical care specialists, nursing experts, and clinical pharmacists, evaluated the content validity, linguistic clarity, and clinical relevance of the questionnaire items."

Paragraph 2.2.1: The sentence about the sample size could be simplified. For example, "A sample size of at least 330 to 660 participants was estimated to be sufficient for factor analysis."

Paragraph 2.2.2: The sentence about the online recruitment could be rephrased to: "Online recruitment advertisements and questionnaire links were disseminated through WeChat and WenJuanXing to invite voluntary participation from nurses at selected hospitals."

Paragraph 2.2.3: The explanation of Cronbach's alpha and ICC could be simplified. For example, "Cronbach's alpha was used to assess the internal consistency of the questionnaire. A value greater than 0.7 indicates good internal consistency."

Statistical Analysis:

The statistical results are well-presented, but for a broader audience, briefly explain the significance of these indices (e.g., Cronbach's alpha, EFA, and CFA).

Results

pecific Suggestions:

Paragraph 3.1: Consider rephrasing the sentence about the Likert scale to: "The Knowledge dimension used a 5-point Likert scale ranging from 'Not at all familiar' to 'Very familiar.' The Attitudes dimension used a 5-point Likert scale ranging from 'Strongly disagree' to 'Strongly agree.' The Practices dimension used a 5-point Likert scale ranging from 'Never' to 'Always.'"

Paragraph 3.2: Consider rephrasing the sentence about the content validity to: "The questionnaire demonstrated excellent content validity, with an item-level Content Validity Index (I-CVI) ranging from 0.91 to 1.00 and a scale-level Content Validity Index/average (S-CVI/Ave) of 0.98."

Paragraph 3.3: Consider breaking up the long sentence about the participants' characteristics into shorter sentences.

Paragraph 3.4: Consider rephrasing the sentence about the Cronbach's alpha to: "The questionnaire demonstrated excellent internal consistency, with a Cronbach's alpha coefficient of 0.96 for the overall scale and 0.96 to 0.98 for the individual dimensions."

Paragraph 3.5: Consider rephrasing the sentence about the factor analysis to: "Exploratory factor analysis (EFA) was conducted on 239 samples. The Kaiser-Meyer-Olkin (KMO) measure of sampling adequacy was 0.945, indicating that the data were suitable for factor analysis. Bartlett's test of sphericity was significant (p < 0.001), further supporting the factor analysis

Discussion

Comment 1

You have clearly stated the objective of developing and testing a questionnaire. To strengthen this, emphasize the significance of understanding clinical nurses' knowledge, attitudes, and practices regarding vasoactive agent infusions. For example:

Comment 2:

You mention that this study is the first of its kind in this area. Consider elaborating on why this gap existed and how this research contributes uniquely.

Comment 3:

The link between the questionnaire design and the KAP theory is mentioned but could be elaborated upon. For example, discuss how each component of the theory is operationalized in the questionnaire items.

Comment 4:

You discuss the benefits of the questionnaire for administrators and researchers. Strengthen this by providing specific examples of potential initiatives.

Comment 5:

You’ve identified the limitations well. To improve, suggest practical steps to address these limitations in future research.

Comment 6:

The mention of intelligent technology and organizational safety climate is insightful. Consider expanding this by hypothesizing how these factors might interplay with the KAP dimensions.

Comment 7:

In the conclusion, emphasize the broader implications and potential applications of the questionnaire.

Reviewer #3: Dear author, I congratulate you on your work. I believe your article will provide a valuable contribution to nursing services. The validity and reliability study of the scale appears to be extremely robust. However, if you could validate the scale's results with the outcomes of other studies, we might better observe the impact of your scale.

6. PLOS authors have the option to publish the peer review history of their article (what does this mean? ). If published, this will include your full peer review and any attached files.

**Do you want your identity to be public for this peer review?** For information about this choice, including consent withdrawal, please see our Privacy Policy .

Reviewer #1: **Yes: ** Abdullahi Ibrahim

Reviewer #2: No

Reviewer #3: No

---

## [Author Response · Author response to Decision Letter 1]

10 Mar 2025

Thank you very much for your thoughtful comments and suggestions on our manuscript. We have carefully revised the manuscript in response to your feedback. Additionally, we have provided a point-by-point response to the reviewers' comments, outlining the specific revisions made and have submitted the revision response to the system.

---

## [Decision Letter · Decision Letter 1]

30 Mar 2025

Knowledge, Attitude, and Practice of Vasoactive Agents Infusions Development and Psychometric Properties of a Questionnaire with Chinese Clinical Nurses

PONE-D-24-41867R1

Dear Dr. Ai,

We’re pleased to inform you that your manuscript has been judged scientifically suitable for publication and will be formally accepted for publication once it meets all outstanding technical requirements.

Kind regards,

Elsayed Abdelkreem, MD, PhD

Academic Editor

PLOS ONE

Additional Editor Comments (optional):

Reviewers' comments:

Reviewer's Responses to Questions

**Comments to the Author**

1. If the authors have adequately addressed your comments raised in a previous round of review and you feel that this manuscript is now acceptable for publication, you may indicate that here to bypass the “Comments to the Author” section, enter your conflict of interest statement in the “Confidential to Editor” section, and submit your "Accept" recommendation.

Reviewer #2: All comments have been addressed

2. Is the manuscript technically sound, and do the data support the conclusions?

Reviewer #2: Yes

3. Has the statistical analysis been performed appropriately and rigorously? 

Reviewer #2: Yes

4. Have the authors made all data underlying the findings in their manuscript fully available?

Reviewer #2: Yes

5. Is the manuscript presented in an intelligible fashion and written in standard English?

Reviewer #2: Yes

6. Review Comments to the Author

Reviewer #2: Accepted as is - no revisions required.I appreciate the opportunity to provide feedback and my comments regarding the manuscript were made.

7. PLOS authors have the option to publish the peer review history of their article (what does this mean? ). If published, this will include your full peer review and any attached files.

**Do you want your identity to be public for this peer review?** For information about this choice, including consent withdrawal, please see our Privacy Policy .

Reviewer #2: No

---

## [Editor Report · Acceptance letter]

PONE-D-24-41867R1

PLOS ONE

Dear Dr. Ai,

I'm pleased to inform you that your manuscript has been deemed suitable for publication in PLOS ONE. Congratulations! Your manuscript is now being handed over to our production team.

Kind regards,

on behalf of

Dr. Elsayed Abdelkreem

Academic Editor

PLOS ONE